# The Effect of SARS-CoV-2 Spike Protein RBD-Epitope on Immunometabolic State and Functional Performance of Cultured Primary Cardiomyocytes Subjected to Hypoxia and Reoxygenation

**DOI:** 10.3390/ijms242316554

**Published:** 2023-11-21

**Authors:** Vytenis Keturakis, Deimantė Narauskaitė, Zbigniev Balion, Dovydas Gečys, Gabrielė Kulkovienė, Milda Kairytė, Ineta Žukauskaitė, Rimantas Benetis, Edgaras Stankevičius, Aistė Jekabsone

**Affiliations:** 1Preclinical Research Laboratory for Medicinal Products, Institute of Cardiology, Lithuanian University of Health Sciences, 50161 Kaunas, Lithuania; vytenis.keturakis@lsmu.lt (V.K.);; 2Department of Heart, Thoracic and Vascular Surgery, Medicine Faculty, Medical Academy, Lithuanian University of Health Sciences, 50103 Kaunas, Lithuania; 3Laboratory of Molecular Cardiology, Institute of Cardiology, Lithuanian University of Health Sciences, 50103 Kaunas, Lithuania; 4Institute of Pharmaceutical Technologies, Faculty of Pharmacy, Lithuanian University of Health Sciences, 50166 Kaunas, Lithuania; 5Department of Drug Chemistry, Faculty of Pharmacy, Lithuanian University of Health Sciences, 50166 Kaunas, Lithuania; 6Institute of Physiology and Pharmacology, Lithuanian University of Health Sciences, 44307 Kaunas, Lithuania

**Keywords:** cardiomyocytes, SARS-CoV-2 infection, COVID-19, mitochondrial respiration, contractions, hypoxia–reoxygenation, oxidative stress

## Abstract

Cardio complications such as arrhythmias and myocardial damage are common in COVID-19 patients. SARS-CoV-2 interacts with the cardiovascular system primarily via the ACE2 receptor. Cardiomyocyte damage in SARS-CoV-2 infection may stem from inflammation, hypoxia–reoxygenation injury, and direct toxicity; however, the precise mechanisms are unclear. In this study, we simulated hypoxia–reoxygenation conditions commonly seen in SARS-CoV-2-infected patients and studied the impact of the SARS-CoV-2 spike protein RBD-epitope on primary rat cardiomyocytes to gain insight into the potential mechanisms underlying COVID-19-related cardiac complications. Cell metabolic activity was evaluated with PrestoBlue^TM^. Gene expression of proinflammatory markers was measured by qRT-PCR and their secretion was quantified by Luminex assay. Cardiomyocyte contractility was analysed using the Myocyter plugin of ImageJ. Mitochondrial respiration was determined through Seahorse Mito Stress Test. In hypoxia–reoxygenation conditions, treatment of the SARS-CoV-2 spike RBD-epitope reduced the metabolic activity of primary cardiomyocytes, upregulated *Il1β* and *Cxcl1* expression, and elevated GM-CSF and CCL2 cytokines secretion. Contraction time increased, while amplitude and beating frequency decreased. Acute treatment with a virus RBD-epitope inhibited mitochondrial respiration and lowered ATP production. Under ischaemia-reperfusion, the SARS-CoV-2 RBD-epitope induces cardiomyocyte injury linked to impaired mitochondrial activity.

## 1. Introduction

Severe acute respiratory syndrome coronavirus 2 (SARS-CoV-2) caused a global pandemic of coronavirus disease 2019 (COVID-19). This is a novel coronavirus from a large family of single positive-stranded, enveloped RNA viruses. It can be transmitted person-to-person through secretions from respiratory droplets or contact with contaminated surfaces [1]. SARS-CoV-2 infection severity and lethality increased with age and in people with concomitant pathologies—cardiac, respiratory, renal, hepatic diseases, diabetes, or obesity. Severe COVID-19 is characterised by extensive lung impairment, which can lead to respiratory failure shock, and death. While respiratory symptoms continue to be the primary characteristics of COVID-19, emerging data suggest that the virus can also affect the cardiovascular system, leading to severe complications and potential long-term effects on heart health. Common complications of COVID-19 are arrhythmias, cardiomyopathy, thromboembolism, pulmonary embolism, disseminated intravascular coagulation, haemorrhage and arterial clot formation, sepsis, shock, and multi-organ failure [2].

The viral life cycle consists of several steps, namely, entry, translation, replication, assembly, and egress [3]. Most evidence points towards angiotensin-converting enzyme 2 (ACE2), being the primary receptor for SARS-CoV-2 entry into the cell. Infection by SARS-CoV-2 commences when transmembrane protease serine II (TMPRSS2) cleaves the virus spike proteins, exposing its receptor binding domain (RBD), which then binds to ACE2 and initiates virus endocytosis within the cell. Although ACE2 is primarily expressed in pulmonary alveolar type 2 progenitor (AT2) and respiratory epithelial cells, it is also present in myocardial, ileum, and oesophagus tissues, as well as some kidney cells, with low expression in immune cells [4]. Once inside a cell, viral RNA and proteins localise to mitochondria. It is notable that the signalling pathways related to inflammation are highly regulated by mitochondria [4,5,6]. There are studies that show that, by inducing mitochondrial dysfunction and oxidative stress, SARS-CoV-2 may initiate a feedback loop that promotes a chronic state of inflammation and endothelial damage [7,8].

According to a study by Ashraf, a hypothesis was made that SARS-CoV-2 enters the cell after binding to the ACE2 receptor expressed on an endothelial cell of a blood vessel. The virus then enables endothelial dysfunction, which causes vasoconstriction and, subsequently, ischaemia. Deprived of oxygen, the organ switches to anaerobic respiration, which produces less ATP, creates an acidic environment, and can impair Na^+^/K^+^—ATPase and Ca^2+^—ATPase pumps. Unable to remove sodium and calcium ions and water molecules, cells begin to swell. In response to ischaemia, reperfusion occurs in which excessive reactive oxygen species are generated, and a cytokine storm is initiated. Monocytes present in the bloodstream enter the injured tissue as macrophages to start the process of phagocytosis. However, due to oxidative stress, the macrophages are unable to perform their function properly. The high intensity of inflammation results in organ damage [7]. In COVID-19 patients, ischaemia-reperfusion injury may contribute to myocardial damage, myocarditis, and other cardiac complications [9,10,11,12]. It can also lead to a dysregulated immune response, including producing proinflammatory cytokines, the hallmark of severe COVID-19 cases [11,13].

Cardiomyocytes are the main heart cells with expressed ACE2 receptors. If these cells are damaged by the virus, it could lead to serious heart problems such as impaired contractile function, disrupted electrical activity, and an imbalance of the renin-angiotensin–aldosterone system, which can cause damage to the heart muscle [14]. However, there have not been enough studies to fully understand how SARS-CoV-2 affects the cardiovascular system. To gain a better understanding, we conducted a study on rat cardiomyocytes using a virus RBD epitope for infection modulation. We simulated oxygen deprivation and reoxygenation to provide insight into the potential mechanisms of COVID-19-related heart complications, focusing on cellular functionality and mitochondrial respiration. Our findings suggested that infected primary rat cardiomyocytes suffered from acute cardiac insufficiency, followed by increased inflammation and impaired mitochondrial function due to mitochondrial damage.

## 2. Results

Severe cases of COVID-19 are often linked to cardiac complications. One of the possible causes of this is respiratory distress syndrome caused by coronaviruses, leading to breathing difficulties, damage to blood vessels, disrupted blood flow, and reduced oxygen intake. As a result, organs may not receive enough oxygen, causing hypoxia. The body then tries to compensate for it by restoring oxygen levels, but this process can lead to oxidative stress and organ damage. Furthermore, since coronaviruses primarily enter cells through ACE2 receptors, which are also found in cardiomyocytes, direct exposure to the virus may contribute to heart complications. In addition, it is not clear what damage is caused by viral signalling via ACE2 receptors after spike protein binding and what is induced by viral infection and particle replication. To study the impact of SARS-CoV-2 receptor binding on heart damage, we recreated hypoxia–reoxygenation conditions commonly seen in individuals with COVID-19 and tested the SARS-CoV-2 epitope (SCoV2-epitope) effects on rat cardiomyocytes.

### 2.1. SARS-CoV-2 Reduces Primary Cardiomyocyte Metabolic Activity

Initially, a preliminary study was conducted to analyse cell metabolic activity under hypoxia–reoxygenation conditions using several SARS-CoV-2 spike protein RBD epitopes. Only one RBD epitope, epitope 370–394, exhibited significant changes and was further used for experiments.

To determine the effective concentration of SCoV2-epitope treatment, a dose-dependent study was conducted using the PrestoBlue^TM^ assay. The results showed that cell metabolic activity was significantly altered by treatment with the SCoV2-epitope at doses of 100 ng/mL and 500 ng/mL (Appendix B, Figure A2). Since the lowest significant dose was found to be 100 ng/mL, all subsequent experiments were conducted using this dose.

Later, PrestoBlue^TM^ assay was performed to evaluate cell metabolic activity after SCoV2-epitope treatment in normoxia and hypoxia–reoxygenation conditions. The acquired results indicated a 17% reduction in the metabolic activity of untreated cells following the cycle of hypoxia and reoxygenation compared to healthy cells (normoxia control) (Figure 1). This result validates our hypoxia and reoxygenation model, as it was established that cells undergo shifts in metabolic requirements during hypoxia, and the effort to restore them during reoxygenation can lead to cellular component damage and impact cell metabolic activity [15].

Direct treatment with SCoV2-epitope significantly did not change cell metabolic activity in normal conditions. However, in the hypoxia–reoxygenation group, the application of SCoV2-epitope treatment resulted in a significant decline in cell metabolic activity of 16% and 33% compared to hypoxia–reoxygenation control cells and healthy cells, respectively. Results suggest that the SCoV2-epitope suppresses metabolic activity of the primary cardiomyocyte culture under hypoxia–reoxygenation conditions.

### 2.2. SARS-CoV-2 Modulates the Immune Response in Primary Rat Cardiomyocytes

#### 2.2.1. Changes in Gene Expression

COVID-19 is often associated with acute inflammatory response characterised by increased levels of cytokines. During severe COVID-19 cases, extremely high cytokine levels described as a cytokine storm are reported. To investigate if cardiomyocytes are stimulated for cytokine production by spike protein RBD epitopes, the expression and secretion of main proinflammatory cytokines associated with severe COVID-19 cases were evaluated.

SCoV2-epitope treatment caused significant upregulation of proinflammatory cytokines in primary cardiomyocytes under normal conditions compared to untreated cells (Figure 2).

Data show that *Il-6*, *Cxcl1*, and *Il-1β* expression was upregulated by 23%, 55%, and 110%, respectively. Hypoxia followed by reoxygenation did not increase *Il-6* levels but significantly induced *Cxcl1* and *Il-1β* expression: *Cxcl1* was found to be upregulated by 150%, while *Il-1β* expression was amplified by 330%.

SCoV2-epitope tended to elevate *NF-kβ1* and *TNF* expression in both normoxia and hypoxia–reoxygenation conditions; however, the differences were not significant. In addition, no significant differences in gene expression changes were observed between normoxia and hypoxia–reoxygenation conditions.

#### 2.2.2. Changes in Cytokines Secretion

Under hypoxia and reoxygenation conditions, primary cardiomyocytes treated with SCoV2-epitope secreted TNFα, IL-1β, IL-6, GM-CSF, and CCL2 cytokines in their culture medium (Figure 3). Among the cytokines measured, CCL2 exhibited the highest detected levels, whereas IL-8 was not identified (data not presented). Interestingly, the *Cxcl1* gene, coding a rat IL-8, was notably upregulated in both conditions; however, proteins in the culture medium were not detected. It might be that translated cytokine remained in the cell and was not secreted. After exposing cells to SCoV2-epitope under hypoxia–reoxygenation conditions, a significant rise in GM-CSF secretion was noticed compared to the control group of cells that remained untreated in both standard and hypoxia–reoxygenation conditions. GM-CSF contributes to inflammation by promoting the migration of immune cells to infected or damaged tissues [16].

In contrast, SCoV2-epitope treatment significantly reduced IL-6, CCL2, and TNFα cytokine levels in normal and hypoxia–reoxygenation conditions. Notably, *Il-1β* gene expression after SCoV2-epitope treatment was significantly upregulated in both normoxia and hypoxia–reoxygenation conditions; however, no significant difference in secretion of the protein into the medium after epitope treatment was detected. However, higher levels of IL-1β were detected in hypoxia–reoxygenation groups, indicating this cytokine is most likely released not by the virus itself but rather by hypoxia or hypoxia–reoxygenation, which is also caused by the infection. When IL-1β is released in response to infection or tissue damage, it activates other cytokines production, for instance, TNFα and IL-6 [17]. Low levels of IL-1β production might explain a decrease in TNFα and IL-6 cytokines secretion, showing that such cytokines might not achieve their production peak.

Overall, the data indicate that treatment with SCoV2-epitope leads to an increase in the secretion of proinflammatory cytokine GM-CSF while decreasing the secretion of TNF-α, IL-6, and CCL2, which can contribute to the damage of primary cardiomyocytes and potentially lead to cardiac failure.

### 2.3. SARS-CoV-2 Inhibits Cardiomyocyte Functionality

As the main heart cells, cardiomyocytes are responsible for rhythmic and efficient contractions to pump blood throughout the body. In our study, we filmed cardiomyocyte contractions in SCoV2-epitope-treated and -untreated cultures maintained in normoxia and hypoxia–reoxygenation. Obtained videos were processed with the Myocyter macro program on ImageJ software. Cardiomyocytes beating less than 20 times per minute, fibrillating, or not beating were omitted from the evaluation.

In all experimental groups, regular beating was observed. Results showed that in all the observed parameters, such as beating times, systolic contractions, diastolic contractions, overall contraction time, and amplitude, statistically significant changes between control primary cardiomyocyte groups and SCoV2-epitope-infected cardiomyocyte culture groups were observed in both normoxia and hypoxia–reoxygenation conditions (Figure 4).

Findings indicate that after SCoV2-epitope exposure, systolic contraction times in normoxia and hypoxia–reoxygenation conditions increased by 2.5 and 1.2 times, respectively. Diastolic relaxation time after virus RBD epitope treatment was elongated by 2 times in normoxia and 1.2 times in hypoxia–reoxygenation conditions. In these parameters, the interaction with epitopes had stronger effects in normoxia than hypoxia–reoxygenation conditions; however, contraction time values in hypoxia–reoxygenation groups were higher. Overall, contraction time after SCoV2-epitope treatment was raised approximately 1.2 times in both conditions compared to the control culture.

Data demonstrated that SCoV2-epitope reduced primary cardiomyocyte culture contraction frequency compared to untreated cells in normoxia from 14.33 to 10.8 beats per second and in hypoxia–reoxygenation conditions from 4.09 to 3.88 beats per second. In addition, maximum amplitudes of cardiomyocyte contractions after SCoV2-epitope treatment were lowered in both conditions more than 2 times (in normoxia 2.3 times and in hypoxia–reoxygenation 2.4 times).

Research unveiled that SCoV2-epitope-affected primary cardiomyocytes in both normoxia and hypoxia–reoxygenation conditions suffered from acute cardiac insufficiency, which was determined by the decrease of beating time predisposed by the elongation of systolic and diastolic times—systolic and diastolic dysfunction. A sharp decrease in amplitude was also observed, consistent with acute heart failure found in other works [18]. Our findings could be interpreted as the onset of acute heart insufficiency caused by the SCoV2 infection, as observed in studies by Siddiq et al. [19].

### 2.4. SARS-CoV-2 Spike Epitope Effect on Mitochondrial Respiration of Cardiomyocytes

During viral infections, mitochondria play a significant role and act as the first line of defence when it comes to viral inflammation. Viruses can affect mitochondrial function and signalling pathways, influencing the immune response and overall cellular behaviour.

This study evaluated mitochondrial respiration after SCoV2-epitope interaction with primary rat cardiomyocytes in normoxia and hypoxia–reoxygenation conditions. Cells’ energetic activity was evaluated by Seahorse Mito Stress assay. During real-time testing, the oxygen consumption and pH levels in a cell culture medium were measured, allowing the evaluation of mitochondrial respiration (oxygen consumption rate) and glycolysis (extracellular acidification rate), respectively.

The results demonstrated that 24 h treatment of SCoV2-epitope did not cause any significant effects on mitochondrial respiration in primary cardiomyocyte culture compared to untreated cells in both conditions (Figure 5). Moreover, mitochondrial respiration tended to be more active in hypoxia–reoxygenation groups than in normoxia, while glycolysis was inclined to be less active in the same groups, although the differences were insignificant.

Overall, in this study, we evaluated mitochondrial activity after 48 h post-treatment with SCoV2-epitope in both normoxia and hypoxia–reoxygenation conditions, and no significant effects of SCoV2-epitope treatment were observed. It is possible that, over this period, cells activated their compensatory mechanisms. Therefore, another experimental series investigated the acute effect of 3 h SCoV2-epitope treatment on primary cardiomyocyte culture mitochondrial activity. SCoV2-epitope acute treatment significantly inhibited basal mitochondrial respiration and caused a 30% reduction in ATP production (Figure 6a,b). Significant changes in glycolytic activity between experimental groups were not observed (Figure 6c). Thus, SCoV2-epitope acute treatment affects cells’ energy phenotype, shifting it towards more quiescent and glycolytic than in untreated cultures (Figure 6d). Moreover, the SCoV2-epitope-primed cultures showed significantly increased metabolic potential of mitochondrial respiration (Figure 6e). Metabolic potential refers to the capacity of cells to fulfil an energy requirement through both mitochondrial respiration and glycolysis. Together, these results suggest that acute treatment of SCoV2-epitope induces metabolic energy redistribution of primary rat cardiomyocytes from mitochondrial to glycolytic.

## 3. Discussion

COVID-19 is strongly associated with cardiovascular complications, such as arrhythmias, cardiomyopathy, and myocardial damage [20]. It is known that SARS-CoV-2 interacts with and affects the cardiovascular system primarily via the ACE2 receptor [9]. SARS-CoV-2 enters the cell through the ACE2 activation by TMPRSS2 cleaved virus spike protein [9]. Accumulated data in big research centres, like the Mount Sinai Health System, shows that in the SARS-CoV-2-infected patient cohort, more than 10% are at risk of developing cardiac dysfunction during infection [21]. Understanding possible pathological outcomes in SARS-CoV-2 patients is beneficial for future treatment and follow-up strategies to prevent the formation of heart insufficiency in the long term.

Infections of SARS-CoV-2 can cause damage to the heart for various reasons. These include reduced oxygen levels caused by acute respiratory distress syndrome (ARSD), hypoxia–reoxygenation injury [22], thrombi formation [23], direct injury to cardiomyocytes due to viral infection [24], and the inflammatory state of infected patients [25]. One of the outcomes of SARS-CoV-2 injury to cardiomyocytes is compromised contractility. This is a multifactorial outcome that is usually caused by inflammation [25], direct myofibril injury [19], and mitochondrial impairment [26].

In this study, we simulated the hypoxia–reoxygenation conditions that are prevalent among SARS-CoV-2-infected patients, including those who have ischaemic heart disease and observed virus spike protein RBD epitope effects on primary rat cardiomyocytes, to gain a better understanding of the potential mechanisms behind cardiac complications associated with COVID-19. Overall, the study results corresponded with the majority of other studies, which stated that ischaemic injury in the myocardium may have caused outcomes of SARS-CoV-2 infection to be worse, as was disclosed in a meta-analysis by Szarpak et al. [10].

Our study was conducted using rat primary cardiac cell cultures comprising approximately 70% of cardiomyocytes. Based on the literature, for ACE2 expression in rat neonatal cardiac tissue, myocytes account for 96% of mRNA compared to fibroblasts, which represent only 4% of ACE2 mRNA [27]. Thus, cardiomyocytes are more susceptible to COVID-19 infection. Moreover, it is important to note that the SARS-CoV-2 spike RBD (amino acids 333–527) has two parts: the variable receptor binding motif (RBM) (amino acids 438–506) and the conservative core RBD (rest of the region), participating in conformational changes of RBD required for exposing RBM to open state and facilitating direct RBM–ACE2 interactions [28]. Initially, our study tested the effects of active spike RBD-epitope 480–499 (part of the ACE2–RBM interaction) on the primary cardiomyocyte culture, and no significant changes in cell metabolic activity were observed. However, significant effects occurred when the treatment was changed to spike RBD-epitope 370–394, which is the core RBD part and does not interact directly with ACE2. In natural infection conditions, coronavirus spike protein RBD epitope 370–394 contributes to the ACE2 signalling pathway [28]. However, in our study, the primary cardiomyocyte culture appears to show effects that might be independent of ACE2. Interestingly, it is reported that core RBD residues that make limited contact with ACE2 exhibit higher mutation frequencies than those involved in direct binding to ACE2 [29] and are linked to SARS-CoV-2 variants [29], leading to more severe complications and hospitalisation [30]. Thus, our study revealed effects on primary cardiomyocyte culture specifically caused by spike core RBD-epitope 370–394.

Inflammation is one of the most prevalent pathways for cardiomyocyte injury. Our study revealed upregulated *Il-6, Cxcl1,* and *Il1β* proinflammatory gene expression in primary cardiomyocytes after virus RBD-epitope exposure; however, a contradictory reduction of main inflammatory cytokines secretion was observed. This can be explained by the fact that during the acute phase of infection, there is an upsurge in cytokine production and then, through a negative feedback mechanism, they are suppressed [31]. This phenomenon was also noted by Abers et al. [32] when decreased longitudinal trajectories of IL-1α were associated with an increased risk of death in SARS-CoV-2-infected patients and by CCL2 decrease, which was also connected with poor prognosis in a study by Pius-Sadowska et al. [33].

We also found an increase in GM-CSF, which many believe is responsible for the detrimental hyperinflammatory response to COVID-19 [34,35]. In normal conditions, GM-CSF controls the clearance of alveoli but, in cases of severe infection, it can secrete reactive oxygen species and express elevated levels of other proinflammatory cytokines, as noted in a study by Hamilton et al. [16]. Yet the increase of the main proinflammatory factors IL-6 and TNFα was minimal. This might be because the spike S glycoprotein of the coronavirus suppresses the production of IL-6 and TNFα [36]. Moreover, during SARS-CoV-2 infection, IL-6 is internalised by immune cells and binds to the IL6R receptor, activating the production of other cytokines such as GM-CSF, as reported in the KEGG database [37]. Our study results showed *IL-1β* upregulation and protein secretion related to NLRP3 inflammasome activity, which is also found in other studies [38].

In vitro, cardiomyocyte contractions can be used to mimic the changes in myocardial contractions in vivo. We found that the SARS-CoV-2 spike protein RBD epitope significantly affected cardiomyocyte contraction time, amplitude, and beating times. In line with our observations, beating times were significantly lower in virus-affected groups, as noted in previous studies by Siddiq et al. [19]. In addition, our findings indicated a decrease in contraction amplitude and elongated contraction time in cardiomyocytes, indicating acute cardiac impairment. In this case, alterations in contractility during COVID-19 in patients could decrease left ventricular ejection fraction (LVEF), which then leads to heart insufficiency—a complication caused by virus infection damage to cardiomyocytes [19].

In our study, an interesting tendency of mitochondrial respiration activation and glycolysis reduction after hypoxia–reoxygenation treatment was observed, which was also noticed by Fuller et al. [39]. This could be attributed to increased Ca^2+^ deposits during the ischaemic period [40] activating mitochondria respiration and reducing glycolysis in reoxygenation [41]. In addition, other authors reported that muscle cells are less susceptible to hypoxia–reoxygenation stress [42]. Such phenomenon can be explained by the fact that during hypoxia, cells produce a lot of substrates required for respiration as a response to oxygen deprivation and cellular stress; thus, after reoxygenation, cells employ these molecules, resulting in increased respiration.

Like many other viruses, SARS-CoV-2 affects the immunometabolic system, causing a Warburg-like shift [5]. According to bioinformatics analysis [43], the main molecular mechanism of cardiac injury in SARS-CoV-2 infection is mitochondrial dysfunction. Our results revealed that an acute infection by virus epitope suppressed mitochondrial respiration in primary cardiomyocytes, resulting in decreased ATP production and redistributed energy phenotype to more glycolytic. Interestingly, after 48 h post-infection, no difference in virus RBD-epitope-treated and -untreated cardiomyocyte groups was observed. These findings align with other authors’ conclusions [44], that in early infection stages, genes responsible for mitochondrial respiration are downregulated, and in later stages, gene regulation is normalised, but the damage to mitochondria persists, potentially causing severe COVID-19 complications. Disruptions in respiratory chain function and ATP generation can be considered central components of mitochondrial dysfunction [45].

SARS-CoV-2 simulates and alters mitochondrial function using ACE2 regulation and open-reading frames (ORFs) [26]. ORFs, such as ORF3a, can impair mitochondrial homeostasis (biogenesis, fusion, fission, and mitophagy) and function [23]. A recent study by Eirin et al. showed that cardiomyocyte mitochondrial damage could also happen because of hypoxic conditions caused by ARDS and the appearance of microthrombi, which is common in SARS-CoV-2 patients [22]. Several studies [46,47] indicated that patients with already compromised mitochondrial function have a bigger risk factor for mitochondrial damage and subsequently worse outcomes in SARS-CoV-2 infection. Ischaemic heart disease prevalence, as shown by Zhang et al.’s [47] meta-analysis, also confirmed that impaired mitochondrial respiration or a decrease in its ability to produce ATP can lead to worse outcomes in SARS-CoV-2-infected patients.

Cardiomyocytes rely on mitochondrial oxidative phosphorylation to produce the energy needed for contraction. When there is a decrease in the synthesis of ATP by the mitochondria, it can negatively impact the function of the heart. In this study, we found reduced cell metabolic activity after virus spike RBD-epitope 370–394 treatment in hypoxia–reoxygenation conditions, together with suppressed mitochondrial respiration and impaired contractility results, suggesting that cardiomyocyte injury by SARS-CoV-2 is associated with mitochondrial damage, induced in the early stages of infection. Our work might give some insights for future studies, concentrating on mitochondrial dysfunction in SARS-CoV-2-infected patients and focusing on preventing cardiac damage.

## 4. Materials and Methods

### 4.1. Experimental Design

The experimental design of this study is illustrated in Figure 7. At first, primary cardiomyocyte cultures were isolated from rat hearts, cultivated, and primed with 100 ng/mL SARS-CoV-2 spike protein RBD epitope 370–394 (Sigma-Aldrich, St. Louis, MO, USA) for 24 h. Treated groups are referred to as Spike groups. An unprimed primary cardiomyocyte culture was used as a control. Further, both groups were maintained in standard conditions, referred to as normoxia. In parallel, other groups of primed and unprimed cells were kept in a hypoxic chamber for 24 h at 37 °C, 2% oxygen, 5% carbon dioxide, and 92% nitrogen gas. Prior to a transfer to hypoxic conditions, the cell culture medium was replaced with a hypoxic medium (standard growth medium kept in a hypoxic chamber for 24 h). After incubation, the medium changed to the standard growth medium again and then, cells were placed in a cell culture incubator for the next 24 h for reoxygenation. In short, 24 h hypoxia (2% oxygen) was followed by 24 h reoxygenation. Further, these cell groups in hypoxia–reoxygenation conditions were referred to as Control and Spike. Overall, four experimental groups were obtained: two in normoxia and two in hypoxia–reoxygenation conditions. Later, cell viability was assessed. Also, inflammatory cytokines gene expression was evaluated, and protein production was determined in a cell culture medium. In addition, cell contractions and mitochondrial respiration were analysed. During cell functionality assay, adrenaline was applied to stimulate cell activity before the experiment.

### 4.2. Primary Culture of Cardiomyocyte Cells

Primary cardiomyocyte cells were prepared from neonatal rat pups according to a modified protocol from Toraason et al.’s [48] study. All investigation procedures were performed according to the Republic of Lithuania Law on the Care, Keeping, and Use of Animals. The rodents were maintained at the Lithuanian University of Health Sciences animal house in agreement with the ARRIVE guidelines.

For isolation of cardiomyocytes, 4–6-day-old neonatal *Wistar* rats were sacrificed by decapitation using sterile scissors, and the chest was opened to extract the heart. The heart was removed using sterile sharp-end forceps and transferred in cold (+4 °C) HBSS (GIBCO^®^, Life Technologies Limited, Inchinnan, UK), without Ca^2+^ and Mg^2+^ ions, solution. The hearts were dissected: the pericardium and both atria were removed with sterile instruments. The hearts were then cut into smaller pieces and centrifuged at 100× *g* speed for 3 min to separate and remove any remaining blood. The pellet was resuspended in 2.5% trypsin-EDTA (GIBCO^®^, Life Technologies Limited, Inchinnan, UK) solution supplemented with papain solution (10 UA for one heart), and then incubated in the fridge overnight. After incubation, cells were homogenised with Pasteur pipettes. The primary cell suspension was centrifuged at 500× *g* for 5 min, twice. The supernatant was removed and the pellet was resuspended in growth medium—DMEM with GlutaMAX^TM^ (GIBCO^®^, Life Technologies Limited, Inchinnan, UK) supplemented with 10% of foetal bovine serum (FBS) (GIBCO^®^, Life Technologies Limited, Inchinnan, UK) and 1% penicillin–streptomycin (GIBCO^®^, Life Technologies Corporation, Carlsbad, CA, USA). The cell suspension was filtered through a 40 µm pore size strainer. Afterwards, cells were counted using Trypan Blue Solution (Sigma-Aldrich, Gillingham, UK) via a haemocytometer chamber and seeded for experiments in cell culture dishes, covered with 0.1% gelatine, and kept at 37 °C and 5% CO_2_ incubator.

According to morphological assessment by specific actin pattern stained with Alexa Fluor™ 568 Phalloidin (Thermo Fisher Scientific, Cambridge, UK), the isolated primary culture comprised 65–70% cardiomyocytes, 25–30% fibroblasts, and up to 10% endothelial cells. Cardiac muscle fibres are composed of myofibrils that are defined by a homogenous succession of transverse stripes containing repeating individual units. Cytoskeleton from actin forms the scaffold of the cell, as it regulates shape, pattern, and, importantly, structure framework [49]. A representative image of a primary cardiomyocyte is shown in Appendix A Figure A1.

### 4.3. Cell Metabolic Activity Evaluation

Cell metabolic activity was evaluated using PrestoBlue^TM^ Cell Viability Reagent (Invitrogen^TM^, Life Technologies Limited, Cambridge, UK). It is a membrane-permeable, non-toxic, and resazurin-based solution, due to which living cells undergo the reduction process of non-fluorescent resazurin to highly fluorescent resorufin. The conversion rate is directly correlated with cell metabolic activity; therefore, it can be used as a cell health indicator to accurately quantify cell viability. The experiment was performed according to the manufacturer’s protocol. First, cells were seeded to two 96-well plates at a density of 8 × 10^4^ cells/well. One plate was kept at normoxia conditions; the other was transferred to 24 h hypoxia followed by 24 h reoxygenation. The medium in the wells was changed after hypoxia and after normoxia. Cell metabolic activity was evaluated 48 h after the cells were seeded, then 10 µL of PrestoBlue^TM^ reagent was combined with 90 µL of cell culture medium in each well of a 96-well plate. The plate was then stored in the dark at 37 °C for 30 min. After incubation, the medium was collected and put into the non-translucent black plate for fluorescence assessment by multimode plate reader Infinite 200 Pro M Nano Plex (Tecan, Männedorf, Switzerland) at 560 nm (excitation) and 590 nm (emission) wavelengths. The results were expressed as an average percentage of control cells in normoxia ± standard deviation.

### 4.4. Gene Expression Assay

Primary rat cardiomyocyte cells were seeded in a 12-well plate at 2 × 10^5^ cells per well and incubated overnight for cell attachment under standard conditions. On the day of the experiment, cells were treated with SARS-CoV-2 Spike protein epitope for 24 h. For hypoxia–reoxygenation experiments, cells were incubated under hypoxic conditions and then reoxygenated for 24 h. After incubation, the total RNA extraction procedure was performed using PureLink RNA mini kit (Thermo Fisher Scientific, Vilnius, Lithuania) according to the manufacturer’s manual and single-stranded cDNA was synthesised by High-Capacity cDNA Reverse Transcription Kit (Thermo Fisher Scientific, Vilnius, Lithuania). Real-time quantitative PCR using Power SYBR green chemistry (Thermo Fisher Scientific, Vilnius, Lithuania) was performed to evaluate changes in interleukin 6 (*IL-6*), C-X-C motif chemokine ligand 1 (*Cxcl1*), interleukin 1 beta (*IL-1β*), tumour necrosis factor (*TNF*), and nuclear factor kappa beta subunit 1 (*NF-kβ1*) gene expression. The glyceraldehyde 3-phosphate dehydrogenase (*GAPDH*) gene was used as an endogenous control. Changes in relative gene expression were normalised to untreated cells and represented as fold change values. The sequences of PCR primers used are listed in Appendix C Table A1.

### 4.5. Detection of Inflammation Protein Markers

The production of inflammation protein markers was evaluated in cell culture supernatants. Cytokines IL-1β, 6, TNFα, granulocyte-macrophage colony-stimulating factor (GM-CSF), and monocyte chemoattractant protein-1 (CCL2) were quantified using Rat Custom ProcartaPlex Mix&Match 5-Plex Kit (Thermo Fisher Scientific, Vilnius, Lithuania), while IL8 was quantified using Rat ELISA Kit (Abbexa Ltd., Cambridge, UK). All procedures were performed according to the manufacturers’ provided protocols. In general, for multiplex immunoassay, cell culture supernatants were incubated with specific antibodies captured to different microbead regions, and then those complexes were labelled by second antibodies with distinct fluorescent probes and the fluorescence signals that occurred were measured by Luminex^®^ 100/200 xMAP instrument (Luminex Corporation, Austin, TX, USA). The results were analysed by ProcartaPlex Analyst 1.0 software (Affymetrix, Santa Clara, CA, USA). For the ELISA plate, an optical density at 450 nm wavelength was measured by multimode plate reader Infinite 200 Pro M Nano Plex (Tecan, Männedorf, Switzerland).

### 4.6. Assessment of Cell Functionality

For the evaluation of the functionality of cardiomyocytes, their contraction parameters were analysed. Primary cell culture was cultivated for 5–7 days until the first contractions appeared. All four experimental groups remained as previously described in Section 4.1. Prior to the evaluation, cells were stimulated with a 0.1 mM adrenaline (Warszawskie Zaklady Farmaceutyczne Polfa, Warsaw, Poland) solution in a growth medium for 10 min at 37 °C. After incubation, primary cardiomyocyte cultures were investigated with a light microscope Leica DMi1 (Leica Microsystems, Veclar, Germany) at 40-fold magnification to capture single cardiomyocytes. Videos of contractions were recorded using the Leica camera system. The obtained videos were analysed using the Myocyter plugin for ImageJ software (version 1.52a), as described by Grune et al. [50]. Cells with apparent contractions were marked and their amplitude changes over time were plotted in a graph (Figure 8).

The amplitude is artificially divided by thresholds at 10, 20, 50, and 90% and was recalculated for every single recognised contraction. The contractions represent the difference between the local minima and the following maxima. The distance between two subsequent maxima determines the contraction time. We chose to evaluate the contraction time from a 10% threshold to a 90% threshold of the amplitude (both systolic and diastolic) to avoid any excessive deviations. Also, the program allows the measurement of the maximum amplitudes and contraction frequency. Summarised graphical images were created with GraphPad Prism 9.0.0 (Dotmatics, Boston, MA, USA).

### 4.7. Evaluation of Mitochondrial Respiration

The mitochondrial respiration functionality of cardiomyocytes was assessed by Seahorse XFp Analyser (Agilent Technologies, Santa Clara, CA, USA) using commercial reagent kit Seahorse XFp Cell Mito Stress Test Kit (Agilent Technologies, Santa Clara, CA, USA) and following the manufacturer’s instructions. First, cells were seeded at a 5 × 10^4^ cells/well density in Seahorse XFp well plates and cultivated for 4–5 days in a complete growth medium (described in Section 4.2) until 60–90% confluency was observed. Prior to this investigation, cells were treated with 100 ng/mL of SARS-CoV-2 spike protein RBD-epitope for 24 h and kept in 48 h normoxia and 24 h hypoxia followed 24 h reoxygenation. Another experiment was conducted with acute SARS-CoV-2 spike protein RBD-epitope treatment for 3 h, and then cells were kept in a cell culture incubator at 37 °C. In parallel, control samples were tested. Following the incubation period, the wells were rinsed with PBS, and then each well was filled with assay medium and repeatedly incubated at 37 °C for 1 h in a non-CO_2_ incubator (Thermo Fisher Scientific, Austin, TX, USA). The assay medium consisted of Seahorse XF DMEM Medium (Agilent Technologies Inc., Santa Clara, CA, USA) supplemented with 10 mM glucose, 1 mM sodium pyruvate, and 2 mM L-glutamine. Afterwards, a Mito Stress Test was performed. During the test, a few mitochondrial respiration modulators, one by one, were injected into each well. The final concentrations in the well of these substances were 1.5 μM oligomycin, 2 μM carbonyl cyanide 4-phenylhydrazone (FCCP), 0.5 μM antimycin A, and 0.5 μM rotenone. The obtained result values of the oxygen consumption rate (OCR) and extracellular acidification rate (ECAR) were normalised to total cellular protein content by Bradford assay. After reaction with the Bradford reagent (Sigma-Aldrich, Hamburg, Germany), optical density was assessed by multimode plate reader Infinite 200 Pro M Nano Plex (Tecan, Männedorf, Switzerland) at 595 nm wavelength. Individual run reports were generated and analysed by Wawe 2.6.1 software (Agilent Technologies, Santa Clara, CA, USA), while summarised graphical images were created with GraphPad Prism 9.0.0 (Dotmatics, Boston, MA, USA).

### 4.8. Statistical Analysis

The quantitative results are presented as mean ± standard deviation of 3–6 replicates. Gene expression data are represented as fold changes ± standard deviation. The statistical data analysis was performed by applying the ANOVA with LSD post hoc test or non-parametric Mann–Whitney U test. Differences were considered statistically significant when *p* < 0.05. The data and statistical analysis were processed using GraphPad Prism 9.0.0 (Dotmatics, Boston, MA, USA) software.

## Figures and Tables

**Figure 1 ijms-24-16554-f001:**
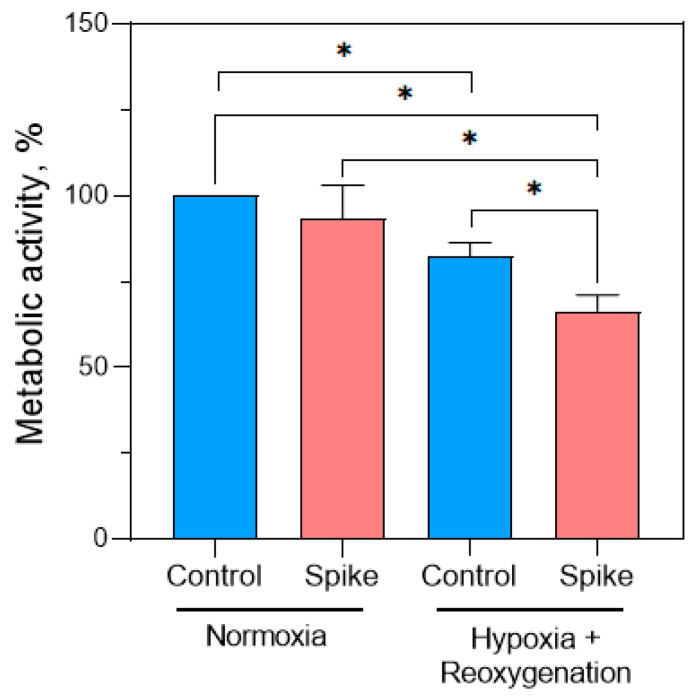
SCoV2-epitope treatment effect on primary cardiomyocyte culture metabolic activity. In normoxia, SCoV2-epitope treatment did not change cell metabolic activity. Treatment followed by hypoxia and reoxygenation conditions significantly reduced the metabolic activity of primary cardiomyocytes. Results are expressed as an average percentage of healthy cells (normoxia control) ± standard deviation. * *p* < 0.05; LSD post hoc test.

**Figure 2 ijms-24-16554-f002:**
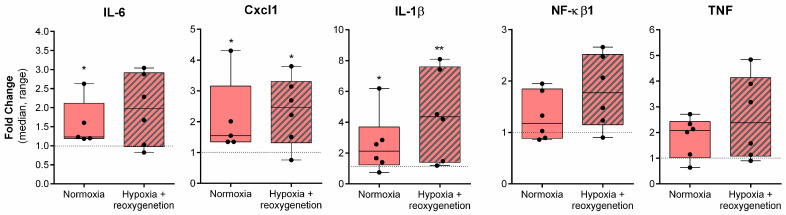
Proinflammatory gene expression changes in rat cardiomyocyte cells after SCoV2-epitope treatment for 24 h. Treatment caused a significant upregulation of *Il-6*, *Cxcl1*, and *Il-1β* under normal conditions. Hypoxia and reoxygenation further induced *Cxcl1* and *Il-1β*, but not *Il-6* expression. Treatment with SCoV2-epitope did not have significant effects on *NF-kβ1* and *TNF* expression. Results are normalised to untreated cells and represented as fold change values. * *p* < 0.05; ** *p* < 0.01; Mann–Whitney U test.

**Figure 3 ijms-24-16554-f003:**
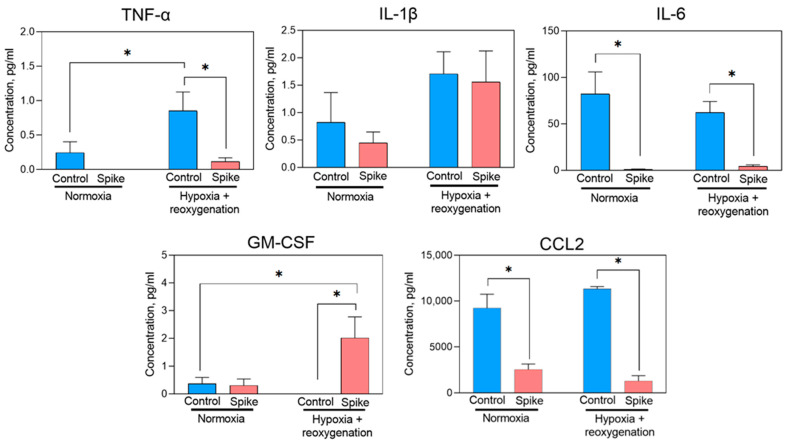
Proinflammatory cytokine secretion by primary cardiomyocytes after treatment of SCoV2-epitope for 24 h. Treatment caused a significant decrease in IL-6 and CCL-2 production in normoxia and hypoxia–reoxygenation conditions. TNFα secretion notably decreased after SCoV2-epitope exposure in both normoxia and hypoxia–reoxygenation conditions. A significant increase in GM-CSF secretion was detected in hypoxia–reoxygenation after the SCoV2-epitope treatment. No significant difference in IL-1β production after the epitope treatment was observed. Results are represented as means ± SEM. * *p* < 0.05; LSD post hoc test.

**Figure 4 ijms-24-16554-f004:**
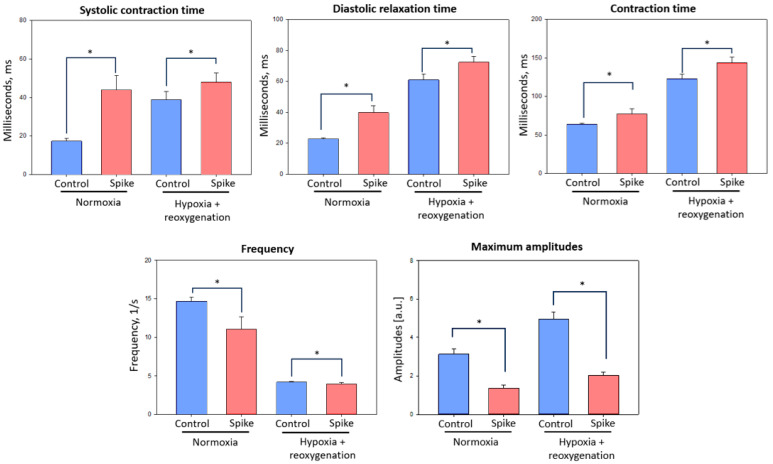
Contraction parameters of primary rat cardiomyocyte culture after SCoV2-epitope exposure for 24 h. Treatment significantly elongated systolic contraction, diastolic relaxation, and overall contraction times in normoxia and hypoxia–reoxygenation conditions. In addition, the treatment significantly decreased frequency and maximal amplitude. Results are presented as corresponding parameters’ means ± standard deviation. * *p* < 0.05; Mann–Whitney U test.

**Figure 5 ijms-24-16554-f005:**
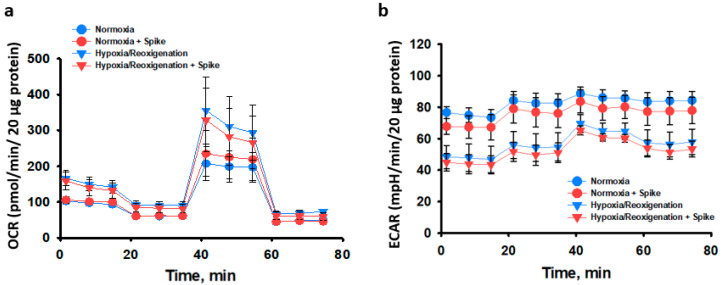
The effect of 24 h SCoV2-epitope treatment on primary cardiomyocyte culture bioenergetics after 48 h post-infection in normoxia and after 24 h hypoxia followed by 24 h reoxygenation. Seahorse Mito Stress Test data. (**a**)—Oxygen consumption rate (OCR) representing the efficiency of mitochondrial respiration. (**b**)—Extracellular acidification rate (ECAR) representing the efficiency of glycolysis. No significant differences between treated and untreated cells were observed. Results are presented as means ± standard deviations of each measurement time point.

**Figure 6 ijms-24-16554-f006:**
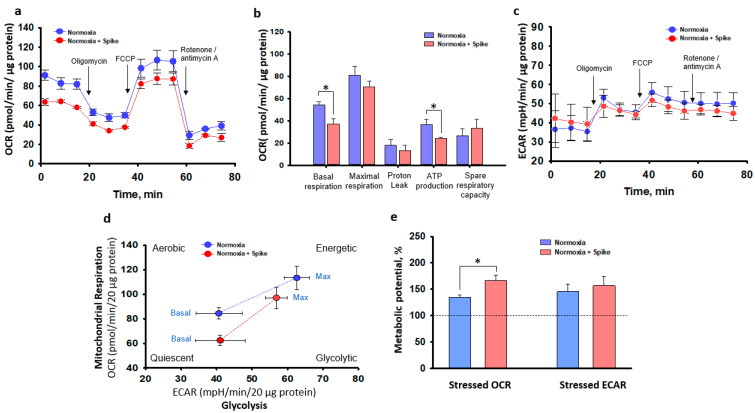
An acute 3 h exposure of SCoV2-epitope effects on primary cardiomyocyte culture bioenergetics in normoxia assessed by Seahorse Mito Stress Test. (**a**)—Oxygen consumption rate (OCR) representing the efficiency of mitochondrial respiration. The first three measurements signify the baseline mitochondrial respiration. The subsequent three measurements, taken after inhibiting ATP synthase with the addition of oligomycin, correspond to oxygen consumption stimulated by proton leak. This is followed by three measurements representing the maximum capacity of mitochondrial oxygen consumption, achieved when the inner mitochondrial membrane is uncoupled using FCCP. Finally, the last three measurements account for oxygen consumption that is not related to mitochondria, occurring when the mitochondrial respiratory chain is hindered by rotenone and antimycin A. Data show that treatment suppressed mitochondrial respiration. (**b**)—Summary data calculated from (**a**) curves. A significant decrease in basal respiration and ATP production was observed after treatment with SCoV2-epitope. (**c**)—Extracellular acidification rate (ECAR) representing the efficiency of glycolysis indicated no significant differences between treated and untreated cells. (**d**)—Cell energy phenotype map showed an energy phenotype shift of treated primary culture. (**e**)—Represents the ability of cells to meet energy demands via mitochondrial respiration and glycolysis. Results are presented as means ± standard deviations. * *p* < 0.05; LSD post hoc test.

**Figure 7 ijms-24-16554-f007:**
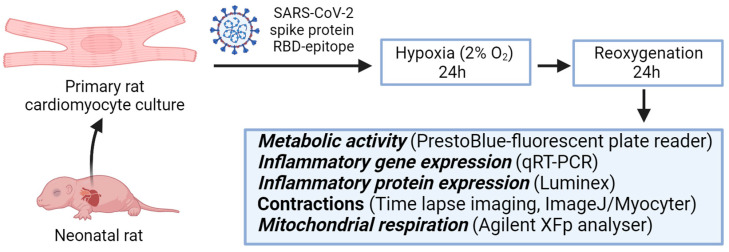
Experimental design scheme. Primary rat cardiomyocytes were treated with SARS-CoV-2 spike protein RBD-epitope, then, for 24 h, were kept in hypoxic conditions followed by 24 h reoxygenation and parallel were kept in normoxia for 48 h. Afterwards, primary cardiomyocyte culture was tested for inflammatory markers, and cell contractions and mitochondrial respiration were analysed.

**Figure 8 ijms-24-16554-f008:**
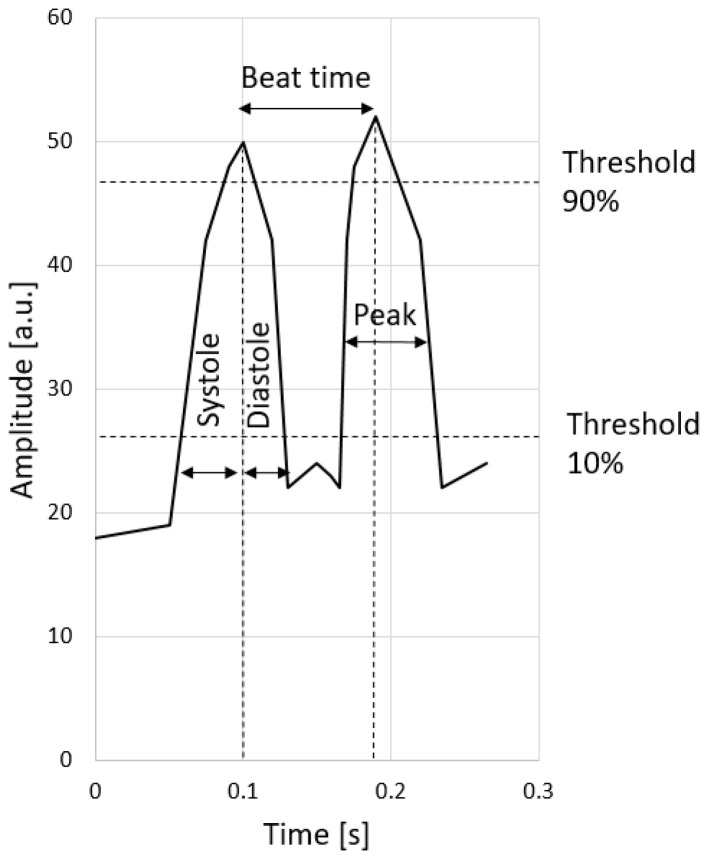
Plotted example of the numerical output of the Myocyter plugin.

## Data Availability

The datasets generated for this study are available on request to the corresponding author.

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
