# Peer review of "The Effect of SARS-CoV-2 Spike Protein RBD-Epitope on Immunometabolic State and Functional Performance of Cultured Primary Cardiomyocytes Subjected to Hypoxia and Reoxygenation"

_ijms, 2023, doi:10.3390/ijms242316554_

Round 1

Reviewer 1 Report

Comments and Suggestions for Authors

In the manuscript entitled “The effect of SARS-CoV-2 spike protein RBD-epitope on immunometabolic state and functional performance of cultured primary cardiomyocytes subjected to hypoxia and reoxygenation”, the authors reported results from their study on the impact of the SARS-CoV-2 spike protein RBD-epitope on primary rat cardiomyocytes under hypoxia-reoxygenation conditions, which including the following findings: (1) In hypoxia-reoxygenation conditions, treatment of SARS-CoV-2 spike RBD-epitope reduced viability of primary cardiomyocytes, upregulated Il1β and Cxcl1 expression, and elevated GM-CSF and CCL2 cytokines secretion; (2) contraction time increased, while amplitude and beating frequency decreased; (3) acute treatment with virus RBD-epitope inhibited mitochondrial respiration and lowered ATP production. This study tried to gain insight into the potential mechanisms underlying COVID-19-related cardiac complications, which is of interest to the research field of SARS-CoV-2. However, the results presented in this manuscript do not support the authors’ conclusions. My comments and concerns are listed below.

(1)    The primary culture of cardiomyocyte cells comprised of cardiomyocytes, fibroblasts, and endothelial cells. The authors did not provide data to show the percentage of each cell type, the expression levels of ACE2 in the three types of cells. Use of the cell mixture makes it difficult to interpretate some of the results, such as increases in the levels of proinflammatory cytokine mRNA. The increased levels are small and cannot be attributed to cardiomyocytes.   

(2)    Results of cytokine secretion is not consistent with those of mRNA expression and indicated that the levels of TNF, IL-6, and CCL2 are significantly reduced in cells treated with SCoV-2 epitope. These results do not support the authors’ conclusion “Overall, the data show that SCoV2-epitope treatment induces inflammatory responses in primary cardiomyocytes and can contribute to damage to the cells, potentially leading to cardiac failure.”

(3)    The PrestoBlueTM assay measures cell metabolic activity and does not directly measure cell viability. Cell metabolic activity is expected to be different under normoxia and hypoxia conditions. It is inappropriate to interpret the data shown in Figure 1 as cell viability and the results do not support the authors’ conclusion, “However, within the hypoxia-reoxygenation group, the application of SCoV2-epitope treatment resulted in a significant decline in cell viability of 16% and 33%, in comparison to hypoxia-reoxygenation control cells and healthy cells, respectively. When comparing the groups affected by the SCoV2-epitope, the epitope causes significantly more damage to cell viability during hypoxic and reoxygenation conditions.”

(4)    The authors used a fixed 100 ng/mL of synthetic SARS-CoV-2 spike protein RBD epitope 370-394 (Sigma-Aldrich, USA) to treat the cells. It is not clear how the 25-mer epitope induced the changes in cardiomyocyte contraction and metabolic activity. No control peptide was used in the study. The authors should perform additional studies to show dose-dependent response.  If it functions via binding to ACE2, the authors should perform ACE2 blockade study.   

Author Response

Dear Reviewer,

Thank you for your insightful comments and valuable suggestions to improve the clarity and experimental evidence presented in our manuscript. We have thoroughly considered your feedback and made the necessary revisions accordingly. For your convenience, we have attached a document that provides a point-by-point response to your comments. Please let us know if you have any further concerns or suggestions.

Best regards, authors

Reviewer 2 Report

Comments and Suggestions for Authors

The authors in the paper “The effect of SARS-CoV-2 spike protein RBD-epitope on immunometabolic state and functional performance of cultured primary cardiomyocytes subjected to hypoxia and reoxygenation” simulate the hypoxia-reoxygenation conditions that are prevalent among SARS-CoV-2 infected patients and they valute virus spike protein RBD epitope effects on primary rat cardiomyocytes. The authors foculased their studies on mitochondrial dysfunction and cardiac damage in SARS CoV-2 infected patients. The article is well presented, the experiments are well designed and clearly presented.

The criticality of the paper is focused on the use of virus RBD-epitope. First, what is your consideration on the real responce of the cells when you simuleted the infection wiyh RBD-epitope in comparison with the complete virus particles? Second, is necessary to consider the Sars CoV2 variants; in this moment the Omicron XBB.1.5 variant, for example, is the most represented among patients. What’s the amminoacids variations per cent in the RBD epitope used for your experiments? It is possible to traslate yours results on the Sars CoV2 variants infections patients?

The correct rational design of the article is crucial for it’s pubblication on international journal of Molecular Science.

Comments on the Quality of English Language

The quality of the languange is good. The article is write clearly.

Author Response

Dear Reviewer,

We would like to express our gratitude to you for taking the time to review our study and providing us with your valuable feedback. Your positive attitude and insightful comments have been greatly appreciated.

We carefully considered your recommendations and have made the necessary revisions to our manuscript, which we believe enhance its clarity and quality. Please find the details in the attachment file.

Best regards, authors.

Round 2

Reviewer 2 Report

Comments and Suggestions for Authors

The manuscript has been improved in terms of clarity and enriched with details that make it more scientifically consistent.

I consider the paper in this new form ready to be published on ijms.

Comments on the Quality of English Language

The authors use a clear and suitable language.